# A retrospective view of the relationship of soluble Fas with anemia and outcomes in chronic kidney disease

**Jessica Felício Andrade**[1☯¤a‡*], **Maria A. Dalboni**[2☯], **Otavio Candido Clemente**[1☯¤a],
**Beatriz Moreira Silva**[1☯¤a], **Barbara Formaggio Domingues**[1☯¤a], **Adelson Marcal Rodrigues**[1☯¤a], **Maria Eugenia Canziani**[1☯¤a], **Abolfazl Zarjou**[3‡],
**Miguel Cendoroglo**[4‡], **Miguel Angelo Goes**[1,5☯¤a¤b‡]

**1** Division of Nephrology, Federal University of São Paulo, São Paulo, Brazil, **2** Department of Research and Graduate, Universidade Nove de Julho/UNINOVE, São Paulo, Brazil, **3** Division of Nephrology, University of Alabama at Birmingham, Birmingham, Alabama, United States of America, **4** Department of Superintendence and Board, Hospital Israelita Albert Einstein, São Paulo, Brazil, **5** Medical School, Faculdade Israelita de Ciências da Saúde Albert Einstein, São Paulo, Brazil

☯ These authors contributed equally to this work.
¤a Current address: Department of Nephrology, Federal University of São Paulo, São Paulo, Brazil
¤b Current address: Medical School, Hospital Israelita Albert Einstein, São Paulo, Brazil
‡ JFA, AZ, MC and MAG also contributed equally to this work.
* andradeljj@yahoo.com

**Data Availability Statement:** All data compiled from the results in excel format files are available from the figshare database (DOI: 10.6084/m9. figshare.21619329).

## Abstract

### Background

Anemia is common in chronic kidney disease (CKD) and is associated with outcomes. In addition, serum soluble Fas (sFas) levels are related to anemia and erythropoietin (EPO) resistance.

### Objectives

Firstly, to compare clinical data and serum levels of sFas, EPO, and pro-inflammatory markers between patients with non-dialytic CKD (NDD-CKD) and healthy subjects. Subsequently, to compare and evaluate the relationship of serum EPO, sFas levels with anemia, and outcomes in patients with NDD-CKD over a long follow-up period.

### Methods

We performed a retrospective study in 58 NDD-CKD patients compared with 20 healthy subjects on complete blood count, kidney function, serum EPO, sFas, and inflammatory markers (CRP, IL- 6, and IFN-γ) at baseline. We then compared the same baseline data between patients with NDD-CKD who evolved to anemia and those who did not have anemia over the follow-up. We also evaluated the frequency of outcomes in patients with CKD with higher sFas levels. Finally, we performed a multivariate analysis of factors associated with CKD anemia.

**Funding:** The authors received no funding for this work.

**Competing interests:** The authors have declared that no competing interests exist.

## Results

There were lower eGFR and Hb but higher serum inflammatory markers, sFas levels, sFas/ eGFR, and EPO/Hb ratios in patients with NDD-CKD. Comparatively, on the other hand, NDD-CKD patients with anemia had lower eGFR but were older, had more diabetes, and had higher sFas/ eGFR, EPO/Hb ratios, and serum levels of IL-6 and sFas than NDD-CKD without anemia for an extended period. In addition, there was an association in a multivariate analysis of diabetes, age, and sFas levels with kidney anemia. Furthermore, there were higher frequencies of outcomes in increased serum sFas levels.

## Conclusion

As an elective risk factor, serum sFas levels, in addition to age and diabetes, were independently associated with kidney anemia for an extended period. Thus, more studies are necessary to analyze the proper relationship of sFas with kidney anemia and its outcomes and therapy in CKD.

## Introduction

Anemia is a common complication of chronic kidney disease (CKD) [1]. It is associated with increased morbidity and mortality and might be related to an accelerated rate of CKD progression [1, 2]. The etiology of anemia in CKD is multifactorial but mainly occurs due to inadequate production of erythropoietin (EPO) [3, 4].

EPO is a glycoprotein composed of 166 aa, and its production occurs mainly in the kidneys. EPO acts as a peptide hormone on its specific receptor (EPOR), distributed in the most diverse tissues [4, 5]. However, EPO's primary function is to act as a growth factor in erythropoiesis since it is essential for producing erythrocytes, thus maintaining tissue oxygenation at optimal levels [5, 6]. But some factors contribute to CKD-related anemia and decreased responsiveness to EPO, such as iron deficiency, inflammation, solutes, and uremic toxins [6–9].

Management of CKD-related anemia consists of erythropoiesis-stimulating agents (ESA) and iron replacement [10–12]. Even so, the blood transfusion required to treat anemia in patients with CKD continues to be widely used, especially in patients with inflammation status who progress to kidney failure with replacement therapy requirements [13, 14]. Moreover, red blood cell (RBC) transfusion presents risks and complications, such as infections and increased sensitization to HLA antigens for kidney transplantation [15, 16].

The soluble form of the Fas receptor (sFas) is produced as an alternative splicing product of CD95/ Fas mRNA or can be cleaved by metalloproteases that lack the transmembrane domain [17, 18]. Thus, sFas may deregulate CD95+-mediated apoptosis [9, 18–20]. Patients with CKD have a decreased kidney clearance of sFas [9, 21]. Previous research has shown a relationship between serum sFas levels with inflammation, cardiovascular disease, and anemia in patients with CKD [18, 22].

In addition, studies reported that serum sFas levels are associated with autoimmunity and hyporesponsiveness to EPO and are predictors of the need for ESA therapy in patients with CKD [9, 17]. Thus, firstly we decided to analyze and compare kidney function, anemia parameters, serum levels of inflammatory markers, EPO, and sFas between patients with non-dialysis dependent CKD (NDD-CKD) and healthy subjects. Afterward, we compared and evaluated the relationship between serum levels of sFas, EPO, and inflammatory markers with anemia

over a long period in patients with NDD-CKD. So, the current study aims to analyze the association between serum sFas levels and anemia for an extended period in NDD-CKD patients.

## Materials and methods

### Patients and data collection

We performed a retrospective study of NDD-CKD patients at the outpatient clinic of the Nephrology Service of the Federal University of São Paulo (UNIFESP) between late 2007 and 2019. In addition, we were able to evaluate data from our database that included patients with NDD-CKD from other publications. [9, 15, 19].

The current study was handled according to the guidelines of the Declaration of Helsinki of 1975, revised in 2013. The Institutional Committee on Human Research, Review and Ethics Board of the Federal University of São Paulo/UNIFESP, SP, Brazil approved the study (396/ 2004, 0108/2010, and 08636819.8.0000.5505/2019).

No individual personal data was revealed or included in our study. All participating volunteers had access to the free and informed consent term. After reading, they signed it, authorizing and approving their participation. Volunteers who agreed to participate in the research could request their withdrawal from the study at any time without interference or discrimination in the treatment they were undergoing. Informed consent included using clinical data and results from anonymously extracted and personally de-identified sera. The researchers stored the patients' data, and it is under total ethical secrecy.

First, all the clinical data of the patients were obtained through the chart medical records. Then, we analyzed the clinical data and serum levels of pro-inflammatory markers, EPO, and sFas, in patients with CKD undergoing conservative treatment. Our data bank contains all the enrolled CKD patients with serum measurements of EPO, sFas, osteo-metabolic and inflammatory markers collected between December 2007 to July 2008. Next, the data from patients with CKD were compared with 20 healthy volunteer individuals. All healthy volunteers had no CKD, diabetes, or hypertension. Thus, the healthy subjects had an eGFR above 60 ml/min/ 1.73 m$^2$ and no marker of kidney damage, such as proteinuria. Therefore, we formed the NDD-CKD and non-CKD groups and evaluated and compared some variables at baseline. So, we performed correlations of the variables in the study of NDD-CKD patients. After, we created two new other subgroups and realized comparisons of parameters at baseline between CKD patients with and without long-term anemia. The analysis of long-term anemia occurred by Hb concentration at the end of follow-up up to 12 years, at the last Hb assessment before initiating kidney replacement therapy, or in evaluating Hb concentration before death among all patients with NDD-CKD. Afterward, we performed multivariate analyzes between NDD-CKD patients using long-term anemia as a dependent variable and variables at baseline as predictor variables.

For all analyses, 126 patients with NDD-CKD filled out the following inclusion criteria: age >18 years and CKD diagnosis for at least three months with CKD stages according to the eGFR between stages 1 to 5, ranging from 11.2 to 91.1 ml/min/1.73. All CKD patients have had proteinuria for at least three months, and blood tests at baselines, such as complete blood count, iron status, kidney function, and serum levels of molecules of interest measured in the study. The exclusion criteria were a change of center and nephrological medical service, referral for kidney transplantation with a living or deceased donor within six months of enrolling in the outpatient clinic, iron depletion, presence of hematological diseases, active bleeding, abnormality in mean corpuscular volume and mean corpuscular hemoglobin, and need for ESA within 30 days after admission to the Nephrology outpatient clinic. In addition, we also excluded patients with a diagnosis of neoplasia, chronic viral diseases such as human

immunodeficiency virus, hepatitis B and C infection, and those who started dialysis as a kidney replacement therapy or died within three months of enrollment in the NDD-CKD outpatient clinic. Thus, we excluded 68 patients from the study (Fig 1).

We collected clinical and demographic data at baseline from patient charts. They included age, sex, active smoking, CKD etiology, body mass index (BMI), complete blood count (hemoglobin concentration, hematocrit, hematological indices, leukocytes, and platelets), iron status, serum albumin, and medications in use, such as a renin-angiotensin system blocker, iron replacement, and use of ESAs. In addition, serum levels of pro-inflammatory markers(C-reactive protein [CRP], interleukin-6 [IL-6], and interferon-gamma [IFN-γ]), EPO, intact parathyroid hormone (iPTH), and sFas, were analyzed from our database of the same patients and with extracted blood sample in the exact period that the clinical tests were analyzed. We also computed the need for ESA, need for red blood cell transfusion, need for dialysis, and mortality as outcomes for up to 12 years of follow-up since admission to the Nephrology outpatient clinic. So the last follow-up and anemia data analysis were for up to 12 years. Or we also got data, such as Hb concentration, before initiating kidney replacement therapy or at the previous assessment before death among all patients with NDD-CKD. We used the sFas/eGFR ratio in correlations and comparisons because that is the molecule evaluated as a uremic retention solute related to clinical outcome.

Anemia was defined as Hb<13.0 g/dl for men and postmenopausal women or Hb<12.0 g/dl for premenopausal women [1, 2, 9, 23]. ESA therapy started when the Hb level decreased to <10.0 g/dl or anemia-induced symptoms were present [9, 23]. The patient's nephrologist determined all drug therapy. The indication of RBC transfusion was by the patient's nephrologists, which we recovered from the patient's medical chart history in whom ESA therapy was ineffective (e.g., erythropoietin resistance or patients with CKD and coronary heart disease with symptoms of anemia such as fatigue, reduced appetite, tiredness, and cognitive impairment) [11, 12, 23].

We calculated the estimated glomerular filtration rate (eGFR) using the CKD Epidemiology Collaboration formula. Thus, we followed the KDIGO criteria to define CKD when eGFR <60 ml/min/1.73m2 or by the presence of proteinuria for at least three consecutive months [24, 25]. Proteinuria was observed in a urine test by a dipstick and confirmed in a urine sample isolated by the colorimetric method (Analisa®, Belo Horizonte, Brazil). We considered proteinuria values greater than 10mg/dl by dipstick urine analysis (Cobas® 6500, Roche Diagnostics GmbH, Germany). Therefore, proteinuria with a value greater than 0.1g/l or when the albumin-to-creatinine ≥30mg/g over three months determined kidney damage [26]. We also used EPO/Hb ratio [18, 27] as an index of response to EPO to perform comparisons at baseline.

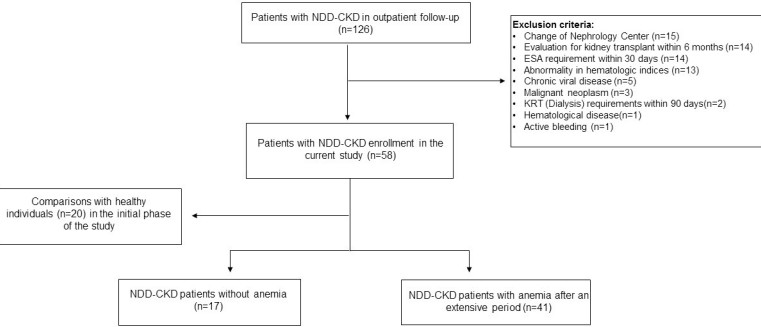

**Fig 1. Workflow diagram of patient selection.** NDD-CKD, non-dialytic chronic kidney disease; ESA, erythropoiesis-stimulating agent; KRT, kidney replacement therapy.

Finally, we also did divide all NDD-CKD patients with higher and lower levels of sFas by the median of serum sFas levels. Thus, we analyzed two other new subgroups of serum sFas levels, higher and lower, to assess and compare the frequency of outcomes (need for ESA, blood transfusion requirement, progression from CKD to dialysis as kidney replacement therapy, and mortality).

## Serum levels

The serum levels of sFas (BD Pharmingen™), EPO (Quantikine Human Erythropoietin Immunoassay R&D systems®, Minneapolis, MN, USA), interleukin-6 (BD OptEIA™ Human IL-6), Interferon-γ (BD OptEIA™ Human IFN-γ; BD Biosciences-Pharmingen, San Diego, CA USA), were measured using enzyme-linked immunosorbent assay, by the manufacturer's instructions. In addition, we analyzed the intact parathyroid hormone (iPTH) and ferritin levels by chemiluminescent microparticle immunoassay with an automatized method (Abbott Laboratories™, IL, USA).

## Statistical analysis

We performed the statistical analyses and graphs in Statistical Package for the Social Sciences (SPSS) version 22 (IBM, Armonk, New York, USA), Excel TM 16.0 (Microsoft, Redmond, Washington, USA), and GraphPad Prism (version 8.0; La Jolla, CA, USA). Numerical data are presented as the mean and standard error to assess the reliability of the norm. In addition, we reported the categorical variables as frequencies and percentages.

In a comparative analysis, we performed the $X^2$ test for frequencies of qualitative variables. The Kolmogorov-Smirnov test was performed to evaluate the normal distribution of continuous data. We carried out serum sFas levels divided by 1000 for statistical analysis. We completed the logarithmic transformation for the data that did not have a normal distribution. Student's t-test was used to compare two groups for variables with normal distribution. Pearson's correlation was performed to evaluate the association between two continuous variables. We carried out the backward stepwise binary logistic regression for admission data using end anemia as a variable response. Predictors variables were included in the multivariate model when they reached $p<0.09$ in the univariate analysis. Regression data were expressed as odds ratios (OR) and 95% confidence intervals (95% CI). It was considered significant at the statistical level of $p<0.05$.

## Ethical issues

There is no individual personal data revealed or included in the current study. Our study was handled according to the guidelines of the Declaration of Helsinki of 1975, revised in 2013. The Institutional Committee on Human Research, Review and Ethics Board of the Federal University of São Paulo/UNIFESP, SP, Brazil approved the study (396/2004, 0108/2010, and 08636819.8.0000.5505/2019). All participating volunteers had access to the written free and informed consent terms. After reading, they signed it, authorizing and approving their participation. Volunteers who agreed to participate in the research could request their withdrawal from the study at any time without interference or discrimination in the treatment they were undergoing. Informed consent included using clinical data and results from anonymously extracted and personally de-identified sera. The researchers stored the patients' data, and it is under total ethical secrecy.

Our study did not interfere with the medical evaluation, request for laboratory tests, or patient management. The patient's nephrologist ordered tests such as complete blood count, kidney function, and electrolytes when appropriate. In addition, the patient's nephrologist

indicated medications, including renin-angiotensin system blockers, ESA, iron and vitamin replacement, and RBCt when necessary.

## Results

The leading causes of CKD were diabetes mellitus and hypertension. In addition, we observed that the NDD-CKD group was older and had higher BMI. Furthermore, the NDD-CKD group showed more elevated serum ferritin, urea, creatinine levels, serum levels of iPTH, sFas, C-reactive protein, IL-6, IFN-γ, sFas/eGFR, EPO/Hb ratio and lower eGFR, Hb, hematocrit (Ht) concentration, platelets, and serum albumin than healthy individuals (Table 1).

We performed correlations of the main variables in the 58 patients with NDD-CKD at baseline. There was a positive correlation between sFas/eGFR and EPO/Hb ratio (Fig 2), sFas levels and EPO/Hb ratio (r = 0.41, p = 0.001), sFas and iPTH levels (r = 0.35, p = 0.04), eGFR and

**Table 1. Comparison of demographic data and clinical parameters at baseline between healthy volunteers and patients with non-dialysis chronic kidney disease.**

|  | non-CKD group (n = 20) | NDD-CKD group (n = 58) | p |
|---|---|---|---|
| **Age (Years)** | 46.7±3.1 | 54.6±1.7 | 0.03 |
| **Gender (n, %)** |  |  |  |
| Male | 10 (50%) | 35 (60.3%) | 0.42 |
| Female | 10 (50%) | 23 (39.7%) |  |
| **BMI (kg/m$^2$)** | 24.7±0.67 | 27.6±0.82 | 0.06 |
| **CKD etiology (n, %)** |  |  | - |
| DM | - | 25 (43.2%) |  |
| Hypertension | - | 17 (29.3%) |  |
| CGN | - | 6 (10.3%) |  |
| Other | - | 10 (17.2%) |  |
| **s-Cr (mg/dl)** | 1.02±0.03 | 2.25±0.12 | <0.001 |
| **eGFR (ml/min/1.73m$^2$)** | 89.7±3.32 | 35.7±2.5 | <0.001 |
| **s-Urea (mg/dl)** | 30.3±1.79 | 74.5±3.62 | <0.001 |
| **Hb (g/dl)** | 14.4±0.25 | 12.8±0.27 | 0.003 |
| **Ht (%)** | 42.5±0.58 | 37.9±0.89 | 0.005 |
| **MCV (fL)** | 91.6±0.57 | 90.8±0.28 | 0.07 |
| **MCH (pg)** | 31.8±0.23 | 31.6±0.14 | 0.61 |
| **Leukocyte (cell/μl)** | 7415±344 | 7656±283 | 0.59 |
| **Platelet (x10$^3$cell/μl)** | 303±9.7 | 219±6.9 | <0.001 |
| **s-Albumin (g/dl)** | 4.6±0.14 | 4.07±0.06 | <0.001 |
| **CRP (mg/dl)*** | 0.24±0.07 | 0.51±0.06 | 0.04 |
| **IL-6 (pg/ml)*** | 3.49±1.29 | 6.64±0.84 | <0.001 |
| **IFN-γ (pg/ml)*** | 2.31±0.76 | 5.70±0.67 | <0.001 |
| **sFas (pg/ml)** | 1136±97 | 2894±172 | <0.001 |
| **sFas/eGFR (pg.ml-1/ml.min-1/1,73m2)*** | 13.2±1.37 | 119.9±13.6 | <0.001 |
| **Serum EPO (mIU/ml)*** | 6.75±0.81 | 10.9±1.42 | 0.09 |
| **EPO/Hb (mIU/ml/mg.dl-1)*** | 4.90±0.58 | 8.76±1.16 | 0.003 |
| **iPTH (pg/ml)** | 40.0±2.09 | 187.4±20.2 | <0.001 |

CKD, chronic kidney disease; BMI, body mass index; DM, diabetes mellitus; CGN, chronic glomerulonephritis; s-Cr, serum creatinine; eGFR, estimated glomerular filtration rate; s-Urea, serum urea; Hb, hemoglobin concentration; Ht, hematocrit concentration; MCV, mean corpuscular volume; MCH, mean corpuscular hemoglobin; s-Albumin, serum albumin; CRP, C-reactive protein; IL-6, interleukin-6; IFN-γ, interferon-gamma; sFas, soluble Fas; iPTH, intact parathyroid hormone; EPO, erythropoietin;

*, after logarithmic transformation for statistical analysis.

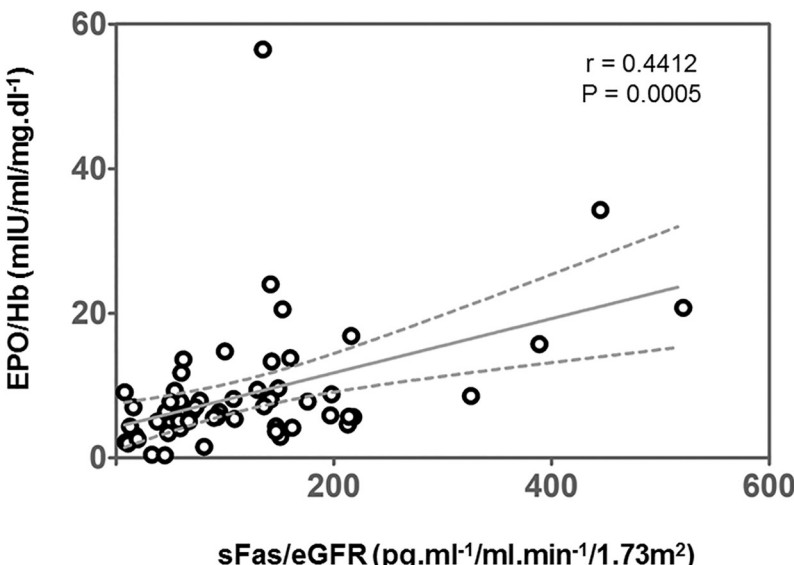

**Fig 2. Correlation between ratios of filtration of soluble Fas and resistance to erythropoietin at baseline (n = 58).**
sFas, serum soluble Fas; eGFR, estimated glomerular filtration rate; EPO, serum erythropoietin; Hb, hemoglobin concentration.

platelets (r = 0.31, p = 0.02), CRP and IL-6 (r = 0.28, p = 0.04), and between sFas and IL-6 (r = 0.26, p = 0.04). We also found negative correlations between eGFR and sFas (r = -0.67, p<0.001), eGFR and IFN-γ (r = -0.42, p = 0.001), sFas and Hb (r = -0.40, p = 0.002), platelets and IL-6 (r = -0.39, p = 0.002), IFN-γ and platelets (r = -0.27, p = 0.04), and between eGFR and IL-6 (r = -0.23, p = 0.03).

Thus, we followed up on the 58 patients with NDD-CKD for 11.3 ±0.7 years. We observed 17 NDD-CKD patients who did not evolve with long-term anemia. Therefore, the long-term hemoglobin concentration was evaluated after 12 years of follow-up, before renal replacement therapy, or even before death. Moreover, 17 patients who were followed up for 12 years did not have long-term anemia but had higher eGFR at baseline. Meanwhile, 41 patients with long-term anemia had lower eGFR at baseline (Table 2), followed up to analyze Hb concentration and outcomes for 10.6±0.4 years.

So, we also observed in Table 2 lower Hb and Ht concentrations at baseline in patients with NDD-CKD in the long-term anemia group than in patients who evolved without anemia after an extensive time. On the other hand, the long-term anemia group had higher creatinine, urea, serum IL-6, sFas levels (Fig 3), EPO/Hb, and sFas/eGFR ratio at baseline than the patients without long-term non-anemia.

Table 3 shows that diabetes mellitus, higher age, and serum levels of sFas at baseline were independently associated with anemia after an extensive period in patients with NDD-CKD.

Furthermore, we observed that the last eGFR (15.5±2.4, 34.6±8.4 ml/min/1.73; p = 0.002) and hemoglobin concentration (10.2±0.3, 14.2±0.6 g/dl; p<0.001) analyzed at the end of follow-up were lower in patients who evolved with anemia in the same period of analysis, that is, long term anemia.

We observed that the median of sFas serum levels was 2294 at baseline. Finally, another division of groups to perform comparisons occurred as follows, higher sFas subgroup when the sFas levels of patients were higher than 2294 pg/ml, and lower sFas for patients with lower serum levels or equal 2294 pg/ml.

**Table 2. Comparison of factors at baseline between chronic kidney disease patients who had anemia and those who did not have anemia at the end of follow-up.**

| | Long-term Non-anemic (n = 17) | Long-term Anemic (n = 41) | p |
|---|---|---|---|
| **Age (Years)** | 44.5±3.4 | 58.8±1.7 | 0.001 |
| **Gender (n, %)** | | | 0.11 |
| Male | 13 (76.5) | 22 (53.6) | |
| Female | 4 (23.5) | 19 (46.4) | |
| **BMI (kg/m$^2$)** | 26.3±0.88 | 28.2±1.10 | 0.19 |
| **DM (n, %)** | 2 (11.8) | 23 (56.1) | 0.002 |
| **Hypertension (n, %)** | 10 (58.8) | 7 (17.1) | 0.001 |
| **s-Cr (mg/dl)** | 1.7±0.13 | 2.49±0.14 | <0.001 |
| **eGFR (ml/min/1.73m$^2$)** | 54.9±5.26 | 27.7±.1.72 | <0.001 |
| **s-Urea (mg/dl)** | 51.9±5.22 | 83.8±3.80 | <0.001 |
| **Hb (g/dl)** | 14.9±0.30 | 12.1.±0.28 | 0.004 |
| **Ht (%)** | 42.6±1.9 | 36.1±1.89 | 0.004 |
| **MCV (fL)** | 90.6±0.50 | 90.7±0.34 | 0.96 |
| **MCH (pg)** | 31.5±0.28 | 31.8±0.16 | 0.47 |
| **Transferrin saturation (%)** | 27.4±2.5 | 27.1±1.7 | 0.91 |
| **Ferritin (µg/L)** | 95.7±18.9 | 89.2±12.3 | 0.77 |
| **s-Albumin (g/dl)** | 4.05±0.12 | 4.08±0.07 | 0.87 |
| **CRP (mg/dl)\*** | 0.49±0.12 | 0.51±0.07 | 0.69 |
| **IL-6 (pg/ml)\*** | 4.02±0.69 | 7.72±1.10 | 0.02 |
| **IFN-γ (pg/ml)\*** | 6.73±2.12 | 5.28±0.37 | 0.31 |
| **sFas (pg/ml)** | 1820±240 | 3339±181 | 0.001 |
| **sFas/eGFR(pg.ml$^{-1}$/ml.min$^{-1}$/1,73m$^2$)\*** | 45.2±9.95 | 150.9±16.6 | <0.001 |
| **Serum EPO (mIU/ml)\*** | 7.62±1.02 | 12.3±1.94 | 0.12 |
| **EPO/Hb (mIU/ml/mg.dl$^{-1}$)\*** | 5.12±0.69 | 10.3±1.56 | 0.04 |
| **iPTH (pg/ml)** | 156±25.3 | 195±26.7 | 0.38 |
| **Use of RAS blocker** | 12 (70.6%) | 35 (85.3) | 0.19 |
| **Active smoking (n, %)** | 6 (35.3) | 18 (43.9) | 0.54 |

BMI, body mass index; DM, diabetes mellitus; s-Cr, serum creatinine; eGFR, estimated glomerular filtration rate; s-Urea, serum urea; Hb, hemoglobin concentration; Ht, hematocrit concentration; MCV, mean corpuscular volume; MCH, mean corpuscular hemoglobin; s-Albumin, serum albumin; CRP,C-reactive protein; IL-6, interleukin-6; IFN-γ, interferon-gamma; sFas, soluble Fas; iPTH, intact parathyroid hormone; EPO, erythropoietin; RAS blocker, renin-angiotensin system blocker;

\*, after logarithmic transformation for statistical analysis.

Eighteen (46.2%) of the thirty-nine patients with NDD-CKD and high serum sFas levels (>2294 pg/ml) required ESA to treat kidney anemia. The CKD patients started ESA therapy with recombinant human erythropoietin 5±2.5 months after admission to the Nephrology outpatient clinic at UNIFESP. Seven (17.9%) of the 39 NDD-CKD patients with higher sFas levels needed RBC transfusion to treat anemia over the 12 years of follow-up. Blood transfusions occurred after 73±12 months of admission to the CKD clinic and enrollment in the study. They received 1.7±0.3 units of red blood cells. No patient with NDD-CKD and low sFas levels (<2294 pg/ml) developed anemia requiring ESA therapy or RBC transfusion.

Moreover, eleven patients (28.2%) of 39 with higher sFas levels evolved with the need for hemodialysis within 26.6±7.2 months. No patient with low serum sFas levels (<2294 pg/ml) at study enrollment became with the need for hemodialysis over 12 years of follow-up. Finally,

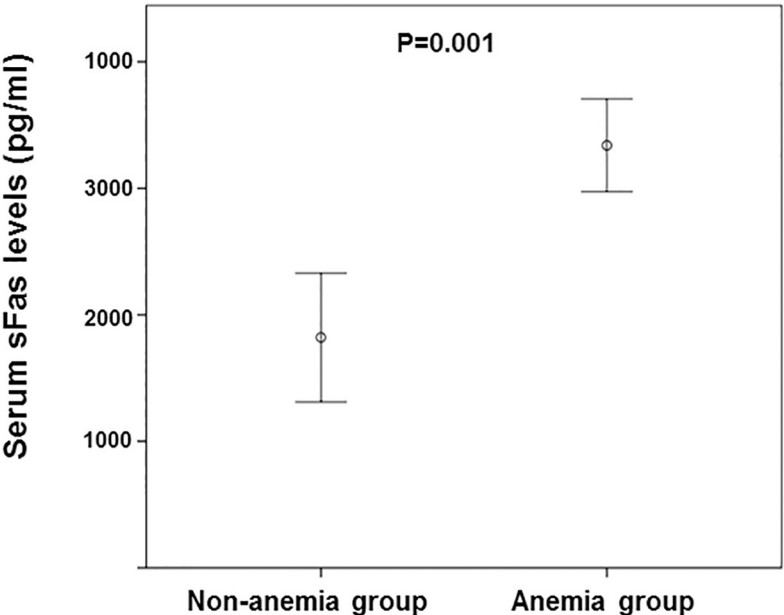

**Fig 3. Comparison of soluble-Fas at baseline between anemic patients (n = 41) and non-anemic patients (n = 17) after follow-up.** sFas, soluble Fas; CKD, chronic kidney disease.

we observed mortality in four NDD-CKD patients with higher levels of sFas (10.3%) in 82.5 +31.6 months over 12 years. At the same follow-up time, one of nineteen (5.3%, p<0.001) patients with lower serum sFas levels showed the same evolution after 89 months.

## Discussion

The relationship of sFas with kidney anemia and outcomes has become evident and scientifically robust in current research. Interestingly, we found higher serum levels of sFas in CKD than in healthy individuals. There was also a positive correlation of sFas with both resistance index to the action of EPO (measured by EPO/Hb) and inflammation (measured by IL-6). Even so, sFas also had a negative correlation with both kidney function (eGFR) and Hb

**Table 3. Binary logistic regression of anemia at the end of the follow-up as the response variable and its predictors at baseline.**

| Long-term Anemia vs. Non-anemia | 95% CI for OR | | | |
|---|---|---|---|---|
| | **OR** | **Lower** | **Upper** | *P value* |
| **DM** | 13.038 | 1.037 | 163.879 | 0.04 |
| **sFas (pg/ml)** | 4.322 | 1.464 | 12.753 | 0.008 |
| **Age (years)** | 1.094 | 1.012 | 1.183 | 0.02 |
| **s-Cr (mg/dl)** | 2.045 | 0.524 | 7.984 | 0.30 |
| **IL-6 (pg/ml)*** | 1.118 | 0.905 | 1.381 | 0.31 |
| **EPO/Hb (mIU/ml/mg.dl⁻¹)*** | 1093 | 0.815 | 1.465 | 0.55 |
| **Hypertension** | 0.785 | 0.101 | 6.111 | 0.82 |

$R^2$ = 0.883; Global model (p = 0.002); OR, odds ratio; 95% CI, 95% confidence interval; DM, diabetes mellitus; sFas, soluble Fas; s-Cr, serum creatinine; EPO, erythropoietin; Hb, hemoglobin;

*, after logarithmic transformation for statistical analysis.

concentration, suggesting an association of serum sFas levels with decreasing kidney function and kidney anemia. We found an independent association in the multivariate analysis of serum sFas levels with long-term kidney anemia. In addition, there was a greater frequency of need for ESA therapy and blood transfusion to treat kidney anemia in the long-term follow-up in patients with more elevated baseline serum sFas levels. More frequent incidences of dialysis and mortality cases also occurred in NDD-CKD patients with higher baseline serum levels. These all were the most critical findings in the current study.

Anemia is frequent in patients with NDD-CKD. For example, Minutolo et al. reported that the prevalence of anemia in NDD-CKD patients was 39.6% in a study from Italy. Furthermore, the incidence and prevalence of anemia in these patients increased in parallel with the CKD stage [27]. In addition, Gill et al. observed in another study with NDD-CKD that the RBC transfusion rate to treat kidney anemia was 2.64/100 person-years. Still, the RBC transfusion rates were higher among those who progressed to kidney failure requiring kidney replacement therapy, with 28.0/100 person-years [28]. Fox et al. previously reported that approximately 20% of patients with NDD-CKD received RBC transfusion to treat kidney anemia [29].

ESA and blood transfusion are typical for treating anemia in patients with CKD and occur with a specific frequency for patients who have CKD progression and are starting kidney replacement therapy or patients who are resistant to the action of ESA, respectively [9, 27–32]. However, red blood cell transfusion increases sensitization to HLA antigens. Therefore, it makes subsequent kidney transplantation more problematic, in addition to the risk of infectious diseases, such as viral hepatitis B and C, as well as HIV, among others, which can increase the severity during immunosuppression [32, 33]. However, we could look further to find molecules associated with kidney anemia, studying serum levels of uremic retention solute related to patients with NDD-CKD. Thus, we could later research new markers of kidney anemia.

Therefore, we undertook the present study to determine the association of serum sFas levels with kidney anemia, ESA requirement, and the need for red cell transfusion in patients with NDD-CKD to obtain more information on the long-term relationship between a uremic retention solute, sFas, and anemia. Moreover, we also investigated the relationship and frequency of sFas with mortality in patients with NDD-CKD.

Thus, the initial phase of the current study compared healthy subjects with patients with CKD-NDD through demographic data, kidney function using estimated GFR [25, 26], complete blood count, and some molecules, such as levels of inflammatory markers, iPTH, and sFas. We found that patients with NDD-CKD had anemia, higher levels of sFas, inflammatory markers, and elevated iPTH levels. However, our data still showed low levels of platelets and serum albumin. In addition, we observed a higher rate of inadequate response to EPO through an elevated EPO/Hb ratio in patients with CKD [34].

Our study reproduced previous findings related to the increase in sFas [9, 18], as well as the lower EPO response [34]; the findings of this work agree with recent studies, which show the increase in inflammatory markers [35] and iPTH [36], but they also offer the low EPO responsiveness [34, 37].

Forty-one patients (70.7%) with NDD-CKD were diagnosed with anemia after an extended period of the current study. Thus, we analyzed and compared clinical factors, serum inflammatory markers, EPO, and sFas levels from the baseline levels between patients with and without anemia within up to 12 years of follow-up. We found that the sFas/eGFR and EPO/Hb ratios at baseline were higher in patients with long-term anemia, suggesting that sFas is associated with EPO hyporesponsiveness. Moreover, serum levels of sFas were higher when there was a lower eGFR in an inverse correlation. Many patients with kidney anemia often have chronic inflammation [38, 39]. Therefore, we found higher serum levels of sFas and pro-inflammatory markers (IL-6, IFN-γ, and CRP) in patients with NDD-CKD than in healthy

subjects in our study. In addition, we observed higher baseline serum IL-6 levels in patients who evolved with long-term anemia. Our findings are in line with recent studies in which patients with CKD have high values of IL-6 [40], IFN-γ [41], and CRP [42].

Inflammatory cytokines directly affect cell differentiation in the erythroid pathway, mediating the induction of cell death, reducing intestinal iron absorption, and increasing iron sequestration by cells of the reticuloendothelial system [38, 39]. Even though, in this study, we did not observe differences in iron status in patients with NDD-CKD who did or did not have long-term anemia.

Some uremic toxins may contribute to anemia in CKD; for example, indoxyl sulfate is a protein-bound uremic toxin that acts in anemia due to impaired EPO synthesis by the kidneys due to a suppression of the EPO gene transcription in a factor hypoxia-inducted factor and promotes suppression of the EPO receptor (EPOR)-AKT pathway [43, 44].

Thus, we sought to study the relationship between sFas and long-term kidney anemia. We also examined the relation of serum sFas with the treatment of kidney anemia, such as the need for ESA therapy and RBC transfusion. Because sFas is a molecule that can be considered a uremic retention solute. The sFas is a molecule of approximately 42.3 kDa and arises from alternative splicing or shearing of the CD95+ receptor. Patients with CKD have elevated serum sFas levels related to inflammatory markers, dysregulation of apoptosis, and outcomes [9, 18, 45, 46]. Furthermore, our group has previously reported lower sFas renal clearance in patients with increased serum levels sFas, which was associated with anemia, hyporesponsiveness to EPO, and increased need for ESA in patients with CKD during six years of follow-up, in addition to inhibiting erythroid cell growth in vitro [9, 18, 21].

The current study found an association between the serum sFas levels at baseline and kidney anemia observed over a long follow-up period. We also found a higher frequency of need for ESA therapy and a need for RBC transfusion over 12 years in patients with NDD-CKD. Furthermore, we report that serum sFas level was independently associated with kidney anemia. We showed that for each increase of 1,000 pg/ml in serum sFas levels, there is a 4.3 times greater chance of the individual developing anemia up to 12 years of follow-up.

Beyond that, during up to 12 years of follow-up, we found a 13 times more chance of developing anemia when the NDD-CKD patient had diabetes mellitus. Other researchers previously reported that hyperglycemia determines hypoxic stress in the kidney interstitium, impairing erythropoietin production, and anemia in diabetes aggravates diabetic kidney disease and cardiovascular disease [47]. Our findings showed an independent association between age and kidney anemia. Yang et al. observed that kidney anemia is frequent and an independent risk factor for decreased kidney function among middle-aged and older people [48]. The current study still had a higher incidence of dialysis and mortality in NDD-CKD patients with higher serum sFas.

Although enthusiastic, the present study has some limitations, such as i) it is a retrospective study, ii) the researchers did not carry out any intervention in the group of patients; iii) the small sample size of patients and healthy volunteers; and finally, iv) a possible selection bias, as we focused on patients with NDD-CKD and serum levels of inflammatory markers and sFas that are associated with anemia and hyporesponsiveness to EPO, with most of the patients from our database already reported [9, 18, 21], and therefore we see that more studies will be necessary to reinforce our findings.

On the other hand, our findings have important implications for understanding the clinical expression when categorizing sFas as a possible marker or even if it is a possible uremic retention solute and, thus, might contribute to the development of new studies that focus on pathophysiological processes, and possibly, the treatment of anemia in patients with CKD, as with newer ESA drugs [49–51].

## Conclusion

In conclusion, this study has documented that patients with NDD-CKD have higher serum levels of sFas, IL-6, IFN-γ, C-reactive protein, iPTH, and EPO/Hb ratio than healthy subjects. In addition, the serum sFas, IL-6 levels, and EPO/Hb ratio at baseline were higher in NDD-CKD patients that evolved with long-term kidney anemia. Moreover, age, diabetes mellitus, and serum sFas levels at baseline were independently associated with anemia over a prolonged follow-up period in patients with CKD. Even so, blood transfusion and ESA requirements to treat anemia during up to 12 years of follow-up were higher in patients with elevated serum sFas levels. However, the present study may point to new paths or tests of another uremic retention solute, sFas, with toxic clinical effects in CKD. Consequently, more clinical and pathophysiological studies are necessary for a possible marker of therapeutic intervention for kidney anemia or a new tool therapy to treat anemia associated with CKD.

## Acknowledgments

We thank all patients, healthy volunteers, professionals, and technicians from the Federal University of São Paulo involved in this work for supporting the study. We thank Professor Nestor Schor (in memoriam) for all his assistance and consideration.

## Author Contributions

**Conceptualization:** Jessica Felício Andrade, Maria A. Dalboni, Miguel Angelo Goes.

**Data curation:** Jessica Felício Andrade, Maria A. Dalboni, Otavio Candido Clemente, Beatriz Moreira Silva, Barbara Formaggio Domingues, Adelson Marcal Rodrigues, Maria Eugenia Canziani, Miguel Angelo Goes.

**Formal analysis:** Jessica Felício Andrade, Otavio Candido Clemente, Beatriz Moreira Silva, Barbara Formaggio Domingues, Adelson Marcal Rodrigues, Maria Eugenia Canziani, Miguel Angelo Goes.

**Investigation:** Jessica Felício Andrade.

**Supervision:** Jessica Felício Andrade, Maria A. Dalboni, Maria Eugenia Canziani, Abolfazl Zarjou, Miguel Cendoroglo, Miguel Angelo Goes.

**Writing – original draft:** Jessica Felício Andrade, Maria Eugenia Canziani, Abolfazl Zarjou, Miguel Cendoroglo, Miguel Angelo Goes.

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
