## [Decision Letter · Decision Letter 0]

3 Mar 2023

PONE-D-23-00605A retrospective view of the relationship of soluble Fas with anemia and outcomes in chronic kidney diseasePLOS ONE

Dear Dr. Andrade,

Thank you for submitting your manuscript to PLOS ONE. After careful consideration, we feel that it has merit but does not fully meet PLOS ONE’s publication criteria as it currently stands. Therefore, we invite you to submit a revised version of the manuscript that addresses the points raised during the review process.

Comments to the authors:1. Respond point-by-point to the reviewers' comments. 2. Check the manuscript for grammatical and syntax errors. 3. Discussion section needs to be rewritten. 

We look forward to receiving your revised manuscript.

Kind regards,

Lakshmi Kannan

Academic Editor

PLOS ONE

Journal Requirements:

Reviewers' comments:

Reviewer's Responses to Questions

**Comments to the Author**

1. Is the manuscript technically sound, and do the data support the conclusions?

Reviewer #1: Yes

Reviewer #2: No

2. Has the statistical analysis been performed appropriately and rigorously? 

Reviewer #1: Yes

Reviewer #2: Yes

3. Have the authors made all data underlying the findings in their manuscript fully available?

Reviewer #1: Yes

Reviewer #2: Yes

4. Is the manuscript presented in an intelligible fashion and written in standard English?

Reviewer #1: Yes

Reviewer #2: No

5. Review Comments to the Author

Reviewer #1: These are my comments to the author:

In Material, Methods section:

- It is a retrospective study for non-dialysis CKD patients presented to outpatient clinics, you should give some details about stages of CKD you are followed

- You followed only 97 patients?

- You should specify the date of sample collections and then the follow up period

- You have used healthy control group; you should give some details about how you chose them

- During follow up period which is 12 years, what about outcomes, how many patients, is any patient started dialysis, is there any patients died, any patients transplanted, also you mentioned the exclusion criteria, is it on initial screening or during follow up period?

- Retrospective study means you depend on patients file review, do you do routine laboratory tests including pro-inflammatory cytokines and sFAS???

Result section:

- In table 1 the number on control group is 20, however, in the flow chart figure, the number is 21m which one is correct?

- The mean eGFR for NDD-CKD was 35.7ml/m±2.5, however the eGFR after 12 years follow up for non-anemic NDD-CKD patients in table 2 was 54.9±5.26, is it worth for CKD patients over the long period of follow up?? It should be lower than at the initial presentation?

- It is better to compare between groups at initial presentation and after follow up period

- There are few language errors like in row 8 after table 3 (seven (17.9%) out of 39, the correct out of

-

Discussion part:

- The discussion section looks more as an introduction part with no more details regarding comparing your study with other previous studies

- There are some few language errors in some words for correction please

Reviewer #2: Comments

1- There are few grammar mistakes, confused sentences, and typos throughout the manuscript. It should be revised carefully.

2- The introduction should be one way or another to be extended to include dissertation about erythropoietin (EPO).

3- References should be updated in both the introduction and the discussion (lack of 2021 &2022 references).

4- Results are well organized, but correlations were not discussed in the discussion part of the manuscript.

5- The discussion is poor and lack explanations of many subtracted points, in addition it should be extended; more other previous studies in the same area of the present study should be used to show that the present results match or do not with that of other previous studies.

6- The conclusion was limited only to diabetes mellitus, age, and higher serum sFas levels although the study includes several other variables such as Hb, serum albumin IL-6, IFN-γ, Serum EPO and iPTH which were not discussed well.

7- There are some weak points in the present study: First, the sample sizes of the patients, and the healthy volunteers who participated in the study were small as proven in the article by the authors and this is againt editorial policies and submission guidelines of PLOS ONE. Also, the researchers did not carry out any intervention in the group of patients and they focused on patients with NDDCKD and serum levels of inflammatory markers (which were not referred to in the discussion) and sFas that are associated with anemia and hyporesponsiveness to EPO, with most of the patients already reported.

6. PLOS authors have the option to publish the peer review history of their article (what does this mean?). If published, this will include your full peer review and any attached files.

Reviewer #1: No

Reviewer #2: No

---

## [Author Response · Author response to Decision Letter 0]

1 May 2023

To the Academic Editor:

1. Respond point-by-point to the reviewers' comments. 

#1Comments, we appreciate the comments of all reviewers, and we have modified all suggested points in detail.

2. Check the manuscript for grammatical and syntax errors. 

#2Comments, we greatly appreciate this comment and reinforce the correction of grammar and syntax errors. 

3. Discussion section needs to be rewritten. 

#3Comments, we’d like to thank you for the comment. We emphasize that several parts of the discussion were modified, aiming to adapt the language, clarify information and further enrich the content. 

Corrections related to journal requirements

All the notes made by the journal were made by the authors, which included: the revision of the formatting, notes added to the topic ‘’ethical issues’’, and the clarification related to the Data Availability statement. 

Comments to the authors:

Reviewer #1:

 In Material, Methods section:

1- It is a retrospective study for non-dialysis CKD patients presented to outpatient clinics, you should give some details about stages of CKD you are followed

1#Comments,

Correspondent author: We thank you a lot for this important observation. Yes, We followed up with 58 patients diagnosed with CKD undergoing conservative treatment between stages 1 to 5 according to the eGFR, which ranged from 11.2 to 91.1 ml/min/1.73, and proteinuria after exclusion criteria. Here eight individuals in stage 1 or 2 who had been healthy volunteers with eGFR >60 but had a marker of kidney damage - proteinuria in late 2007/early 2008 in urine analysis in samples collected and analyzed with an interval of more than three months. So, they are currently our CKD outpatients since the diagnosis. As suggested, we wrote the patients' stages and eGFR in the Materials and Methods section.

2- You followed only 97 patients?

2#Comments,

Correspondent author: Thank you again for this comment. 

In fact, in our database, there were 126 outpatients admitted between December 2007 and July 2008 with CKD under conservative treatment, presenting an evaluation of blood count, iron status, kidney function, and urine I. In parallel, they were enrolled for research in our database, with measurement of serum levels of sFas, EPO, osteo-metabolic, and inflammatory markers. We had not reported these patients before because 29 patients changed centers or were followed up for kidney transplants within six months of registration in our database. As amended, described in the main text and Figure 1 after we've looked it over carefully. Thus, we could follow up with 58 patients with CKD after exclusion criteria over 12 years.

3- You should specify the date of sample collections and then the follow up period.

3#Comments,

Correspondent author: We thank you for this information, and we have edited all method sections to describe better the date of sample collections and the follow-up period.

We also computed the need for ESA, need for red blood cell transfusion, need for dialysis, and mortality as outcomes over 12 years of follow-up since admission to the Nephrology outpatient clinic. So the last follow-up and anemia data analysis were for up to 12 years. Thus, we also got data before initiating kidney replacement therapy or at the previous assessment before death among all patients with NDD-CKD.

4- You have used healthy control group; you should give some details about how you chose them.

4#Comments,

Correspondent author: Thank you for this encouraging comment. We have included more details about the healthy volunteers in the second paragraph in Patients and Data collection.

5- During follow up period which is 12 years, what about outcomes, how many patients, is any patient started dialysis, is there any patients died, any patients transplanted, also you mentioned the exclusion criteria, is it on initial screening or during follow up period?

5#Comments,

Correspondent author: Thank you for this critical observation. We have reported the outcomes in Methods and Results. So, we have computed the need for ESA, need for red blood cell transfusion, need for dialysis, and mortality as outcomes for up to 12 years of follow-up since admission to the Nephrology outpatient clinic. During the 12 years of follow-up, 11 patients required a dialysis program. Unfortunately, there were five non-survivor patients over 12 years of follow-up. 

6- Retrospective study means you depend on patients file review, do you do routine laboratory tests including pro-inflammatory cytokines and sFAS???

6#Comments,

Correspondent author: Thank you for this relevant point. In the current study, we have included patients with iron status, complete blood count, kidney function, and urine analysis and collected that patient data from charts. In addition, we have a database for research with serum analysis of sFas, inflammatory markers, and bone profiles parallel to the charts' clinical data. Prof. Miguel Angelo has been studying serum levels of sFas, and inflammatory markers related to anemia in CKD since his Master's degree. Thus, he stores the database safely and ethically. Therefore, after 12 years of follow-up and observing outcomes, we decided to publish these data with approval by the ethics committee of UNIFESP.

Result section:

7- In table 1 the number on control group is 20, however, in the flow chart figure, the number is 21m which one is correct?

7#Comments,

Correspondent author: Thank you for this critical observation.

We have made a mistake in typing. The correct number is 20 healthy individuals. We corrected it in the figure and the main text.

8- The mean eGFR for NDD-CKD was 35.7ml/m±2.5, however the eGFR after 12 years follow up for non-anemic NDD-CKD patients in table 2 was 54.9±5.26, is it worth for CKD patients over the long period of follow up?? It should be lower than at the initial presentation?

8#Comments,

Correspondent author: Thank you for this critical observation.

We appreciate your intelligent and solid question. We redid all the statistical analysis and looked over the typing again.

In fact, we retitled Table 2 for clarity. Thus, in this table and the results in the main body of the text, we analyzed two more subgroups of 58 patients with CKD for the long-term anemia outcome. Thus, we were able to compare patients with CKD and anemia in the long term and patients with CKD without anemia for a long time. However, baseline eGFR, as expected, was higher in the group without long-term anemia (54.9±5.26 ml/min/1,73).Consequently, we used baseline variables in these comparisons and then performed binary logistic regression with long-term anemia as the dependent variable.

However, these analyses supported our hypothesis, as these two subgroups did not have statistically significant serum EPO levels, although patients with long-standing anemia had higher EPO values. These same patients had higher levels of sFas, sFas/eGFR ratio, and EPO/Hb, indicating that the lower relative production is the main factor of anemia but that some uremic solutes retained in renal dysfunction contribute to anemia in these patients.

9- It is better to compare between groups at initial presentation and after follow up period

 9#Comments

Correspondent author: Again, thanks for your critical opinion. We believe there was a mistake from us, and for better understanding, we changed the title of Table 2 to “Comparison of factors at baseline between chronic kidney disease patients who had anemia and those who did not have anemia at the end of follow-up”. We compared the eGFR values at baseline of patients with CKD-NDD who did not have long-term anemia (54.9±5.26) with those who were long-time anemic (27.7±.72 ), both at the end of the follow-up.

10- There are few language errors like in row 8 after table 3 (seven (17.9%) out of 39, the correct out of

10#Comments,

Correspondent author: Thanks a lot. As suggested, we have made changes for better understanding.

Discussion part:

11- The discussion section looks more as an introduction part with no more details regarding comparing your study with other previous studies

11#Comments,

Correspondent author: Again, thank you for your opinion. We sought to enrich the Discussion section by comparing our findings with recent archives in the literature.

12- There are some few language errors in some words for correction please

12#Comments,

Correspondent author: Thank you. As suggested we try to fix language errors in the Discussion section

Reviewer #2

1- There are few grammar mistakes, confused sentences, and typos throughout the manuscript. It should be revised carefully.

1.2#Comments,

 Correspondent author: Thank you for this comment. As suggested, we carefully proofread the text, removing confusing phrases.

2- The introduction should be one way or another to be extended to include dissertation about erythropoietin (EPO).

 2.2#Comments

Correspondent author: Again, thanks for your critical opinion. As suggested, we have included more concepts about erythropoietin (EPO) in the introduction.

3- References should be updated in both the introduction and the discussion (lack of 2021 &2022 references).

3.2#Comments,

Correspondent author: Many thanks, for your opinion.

As suggested, we have included new references in the Introduction and Discussion section after careful reading and review in line with our writing.

4- Results are well organized, but correlations were not discussed in the discussion part of the manuscript.

4.2#Comments,

 Correspondent author: Thank you again for this comment.

As suggested, we reported the relationship of sFas (the focus of our study) with eGFR and some results of the correlation between sFas and both erythropoietin hyporesponsiveness (EPO/Hb ratio) and inflammation (IL-6) in the Discussion Sector.

5- The discussion is poor and lack explanations of many subtracted points, in addition it should be extended; more other previous studies in the same area of the present study should be used to show that the present results match or do not with that of other previous studies.

5.2#Comments,

Correspondent author: Thank you for this critical observation. As suggested, the discussion has been expanded and matched with other research.

6- The conclusion was limited only to diabetes mellitus, age, and higher serum sFas levels although the study includes several other variables such as Hb, serum albumin IL-6, IFN-γ, Serum EPO and iPTH which were not discussed well.

6.2#Comments,

Correspondent author: Again, thank you for this critical observation. As suggested, we discuss more Hb, IL-6 serum albumin, IFN-γ, serum EPO and iPTH in the Discussion Sector. Thus, as suggested, we have included a few more inflammatory markers in the Discussion and Conclusion.

7- There are some weak points in the present study: First, the sample sizes of the patients, and the healthy volunteers who participated in the study were small as proven in the article by the authors and this is againt editorial policies and submission guidelines of PLOS ONE. Also, the researchers did not carry out any intervention in the group of patients and they focused on patients with NDDCKD and serum levels of inflammatory markers (which were not referred to in the discussion) and sFas that are associated with anemia and hyporesponsiveness to EPO, with most of the patients already reported.

7.2#Comments,

Correspondent author: We appreciate this feedback, critical observation, and opinion.

Indeed, as we reported in the manuscript, our study had many limitations. Our sample size at enrollment was larger by our database, which is stored safely and ethically. However, there were many reasons for exclusion, such as the preferential request for a referral for a kidney transplant in the first six months of outpatient follow-up, even non-survival or need for dialysis, or use of ESA in the first three months of outpatient follow-up. In addition, there were severe comorbidities such as HIV, chronic hepatitis, neoplasms, or even hematological diseases that directly interfered with the anemia of these patients. Likewise, our control group was initially formed by 30 volunteers. All of them were employees from our University environment (not described in the manuscript). Of these, seven were excluded due to incomplete data. Unfortunately, two still died in the pandemic, and one asked to no longer participate in the study.

---

## [Editor Report · Decision Letter 1]

25 May 2023

A retrospective view of the relationship of soluble Fas with anemia and outcomes in chronic kidney disease

PONE-D-23-00605R1

Dear Dr. Andrade,

We’re pleased to inform you that your manuscript has been judged scientifically suitable for publication and will be formally accepted for publication once it meets all outstanding technical requirements.

Kind regards,

Lakshmi Kannan

Academic Editor

PLOS ONE
---

## [Editor Report · Acceptance letter]

22 Jun 2023

PONE-D-23-00605R1 

A retrospective view of the relationship of soluble Fas with anemia and outcomes in chronic kidney disease 

Dear Dr. Andrade:

I'm pleased to inform you that your manuscript has been deemed suitable for publication in PLOS ONE. Congratulations! Your manuscript is now with our production department. 

Kind regards, 

on behalf of

Dr. Lakshmi Kannan 

Academic Editor

PLOS ONE